# Geometries, Electronic Structures, Bonding Properties, and Stability Strategy of Endohedral Metallofullerenes TM@C$_{28}$ (TM = Sc$^-$, Y$^-$, La$^-$, Ti, Zr, Hf, V$^+$, Nb$^+$, Ta$^+$)

**Dong Liu, Yuan Shui and Tao Yang ***

MOE Key Laboratory for Non-Equilibrium Synthesis and Modulation of Condensed Matter, School of Physics, Xi'an Jiaotong University, Xi'an 710049, China; dliu_sd@stu.xjtu.edu.cn (D.L.); f24270216f@stu.xjtu.edu.cn (Y.S.)
* Correspondence: taoyang1@xjtu.edu.cn

**Abstract:** We performed quantum chemical calculations on the geometries, electronic structures, bonding properties, and stability strategy of endohedral metallofullerenes TM@C$_{28}$ (TM = Sc$^-$, Y$^-$, La$^-$, Ti, Zr, Hf, V$^+$, Nb$^+$, Ta$^+$). Our calculations revealed that there are three different lowest-energy structures with C$_{2v}$, C$_{3v}$, and T$_d$ symmetries for TM@C$_{28}$. The HOMO–LUMO gap of all these structures ranges from 1.35 eV to 2.31 eV, in which [V@C$_{28}$]$^+$ has the lowest HOMO–LUMO gap of 1.35 eV. The molecular orbitals are mainly composed of fullerene cage orbitals and slightly encapsulated metal orbitals. The bonding analysis on the metal–cage interactions reveals they are dominated by the Coulomb term $\Delta E_{elstat}$ and the orbital interaction term $\Delta E_{orb}$, in which the orbital interaction term $\Delta E_{orb}$ contributes more than the Coulomb term $\Delta E_{elstat}$. The addition of one or two CF$_3$ groups to [V@C$_{28}$]$^+$ could increase the HOMO–LUMO gap and further increase the stability of [V@C$_{28}$]$^+$.

**Keywords:** fullerenes; quantum-chemical calculations; chemical bonding; stability





## 1. Introduction

Fullerenes, structurally unique closed hollow carbon cages, are composed of exactly 12 pentagons and a certain number of hexagons. Among them, buckminsterfullerene C$_{60}$ stands out as the most symmetrical and chemically inert member [1–3]. The smallest fullerene, C$_{20}$, presents a singular dodecahedral structure but is theoretically predicted to be highly reactive [4]. The reason why these small fullerenes are highly reactive is their high curvature. Thus, there are two main strategies to stabilize these fullerenes. On the one hand, there is the encapsulation of metal inside fullerenes, which gives rise to endohedral metallofullerenes (EMFs), which could lead to a charge transfer from metal to fullerenes and stabilize the high curvature motifs [5]. On the other hand, exohedral derivatization on high curvature motifs could release the curvature directly and make them transform from $sp^2$ hybridization to $sp^3$ hybridization [6,7].

EMFs, which encapsulate metal atoms or clusters within fullerene cages [5,8,9], represent an innovative class of metal–carbon clusters. These EMFs combine the structural merits of fullerene frameworks with the photoelectromagnetic advantages of metals. Particularly, C$_{28}$ has been observed to exhibit one remarkable peak in the domain of supersonic carbon cluster beam experiments [4,10]. However, the isolation of C$_{28}$ has been challenging due to its open-shell electronic nature. EMFs not only improve the stability of reactive fullerenes such as C$_{28}$ [11,12] but also introduce novel functionalities. For instance, the encapsulation of tetravalent metals like titanium (Ti), zirconium (Zr), and uranium (U) culminates in the formation of stable endohedral metallofullerenes, namely Ti@C$_{28}$, Zr@C$_{28}$, and U@C$_{28}$ [4,13–17]. U$^{2+}$@C$_{28}$ has been theoretically predicted to adhere to the 32-electron principle [18], of which central f-block metals like uranium have their s-, p-, d-, and f-type valence shells completely filled and exhibit remarkable stability. U@C$_{28}$ also

follows the 32-electron rule and accommodates two valence electrons within the peripheral cage structure [19].

The synthesis of EMFs involves various experimental methods [20], including electric arc discharge [21], which generates EMFs by vaporizing graphite rods and metal oxides in an inert atmosphere, and laser ablation [22], where a high-power laser vaporizes a carbon-rich target and metal oxides to facilitate EMF formation. Chemical vapor deposition (CVD) methods [23,24], involving the thermal decomposition of hydrocarbon gases on metal catalysts, contribute to fullerene production. Additionally, the surgery approach [25] is often employed to open fullerene cages, put transition metals inside, and close fullerene cages.

The presence of a fullerene can also modify the physical and chemical properties of encapsulated atoms. Recently, one theoretical study showed that in $^7Be@C_{28}$, the electron capture decay rate of $^7Be$ could become slower because of the metal–cage interactions [26]. Wang and Gao revealed that a hyperhydrogenated water species, $H_4O$, can be easily formed using $H_2$ and $H_2O$ if they are put inside small-sized fullerenes like $C_{28}$ as nanoreactors [27]. Spano and coworkers theoretically studied the $Cr@C_{28}$ and found there are two stable sites of Cr metal in $C_{28}$, leading to two different measurable currents in single-molecule devices [28]. The bonding, electron affinity, optical, and magnetic properties of $M@C_{28}$ (M = Ti, Zr, Hf, U) have been investigated theoretically [29]. However, the element effect on the geometries and electronic structures is rarely discussed. Moreover, the chemical bonding between the encapsulated metal and fullerene cage $C_{28}$ has been explored.

In the present study, we studied the geometries, electronic structures, bonding properties, and stability strategy of endohedral metallofullerenes $TM@C_{28}$ (TM = $Sc^-$, $Y^-$, $La^-$, Ti, Zr, Hf, $V^+$, $Nb^+$, $Ta^+$) by using quantum chemical calculations. The chemical bonding between metal and $C_{28}$ was studied using the energy decomposition analysis. The stability strategy of the highly reactive EMFs is discussed using trifluoromethyl groups.

## 2. Computational Methods

The structural search was started from two fullerene cages, including $C_{28}$-$D_2$(1) and $C_{28}$-$T_d$(2). The positions of transition metal atoms were generated randomly within a geometric sphere centered on the fullerene cages and with a radius smaller than that of the fullerene cage. In total, over 1000 initial structures were randomly generated. Clustering of these structures was conducted using the ISOSTAT module within the Molclus program [30] with two criteria: (a) an energy difference of less than 0.25 kcal/mol and (b) a geometric deviation of less than 0.1 Å. The geometric deviation was defined by calculating a distance matrix based on the atomic coordinates of the structures and converting its nondiagonal elements into a one-dimensional array. The calculations start from the semiempirical GFN-xTB method [31,32] and follow density functional theory (DFT) calculations at the PBE-D3(BJ)/def2-SVP level [33]. The final optimizations were performed at the PBE-D3(BJ)/def2-TZVPPD level [34–41]. The wave function stability and frequency analysis were subsequently verified, confirming these structures as local minima on the potential energy surface. The atomic charge distribution was computed using natural bond orbital (NBO7.0) [42] at the PBE-D3(BJ)/def2-TZVPPD level to gain electron distribution and bonding interactions. NBO analysis offers a comprehensive view of molecular bonding by translating electron densities into more chemically intuitive patterns like bonds and lone pairs, thus aiding in the understanding of intrinsic molecular interactions. All these DFT calculations were performed using Gaussian 16 software [43].

The energy decomposition analysis (EDA) developed by Ziegler and Rauk [44] is one of the most powerful methods to study the nature of chemical bonds. Here, the bonding nature between the metal dimer and fullerene cage was studied via EDA. Scalar relativistic effects and zeroth-order regular approximation (ZORA) [45] were considered. In EDA calculations, the molecule AB divides into two fragments, A and B, and the kernel is put on the intrinsic interaction between fragments that hold a specified electronic reference state in the frozen geometry [46,47]. The EDA approach decomposes the instantaneous

interaction energy $\Delta E_{int}$ of the A–B bond into four energy terms, of which the first three are the main components:

$$\Delta E_{int} = \Delta E_{elstat} + \Delta E_{Pauli} + \Delta E_{orb} + \Delta E_{disp} \tag{1}$$

The term $\Delta E_{elstat}$ is the *quasi*-classical electrostatic interaction between the unperturbed charge distributions of the prepared atoms and is usually an attractive term. The Pauli repulsion $\Delta E_{Pauli}$ is the energy change in the transformation from the superposition of the unperturbed electron densities $\rho_A + \rho_B$ of the isolated fragments to the wave function $\Psi^0 = N\hat{A}[\Psi_A \Psi_B]$, which properly obeys the Pauli principle through explicit antisymmetrization (Â operator) and renormalization (N = constant) of the product wavefunction. $\Delta E_{Pauli}$ comprises the destabilizing interactions between electrons of the same spin on either fragment. The orbital interaction $\Delta E_{orb}$ derives from charge transfer, electron-pair bonding, and polarization effects. The EDA calculations were performed with the ADF2019 [48,49] package using the BP86 [50,51] functional with Grimme's dispersion correction D3(BJ) [33] and TZ2P+ basis set [52]. The zeroth-order regular approximation (ZORA) Hamiltonian was adopted to include the relativistic effect [45]. All these EDA calculations were performed based on the PBE-D3(BJ)/def2-TZVPPD-optimized structures.

## 3. Results and Discussion

There are two isomers of $C_{28}$, including $D_2(1)$ and $T_d(2)$. Our calculations show that all these metals prefer to be encapsulated inside the $T_d(2)$, giving rise to TM@$T_d(2)$-$C_{28}$. Figure 1 presents the calculated structural models of optimized TM@$T_d(2)$-$C_{28}$ (TM = Sc$^-$, Y$^-$, La$^-$, Ti, Zr, Hf, V$^+$, Nb$^+$, Ta$^+$). All these structures have a singlet as the ground state. However, the optimized structures are quite different for the different metals and slightly depend on the charge state. In the case of [Y@$C_{28}$]$^-$, [La@$C_{28}$]$^-$, and [Hf@$C_{28}$], the metal could stay in the center of the fullerene cage, leading to the $T_d$ symmetry of the whole endohedral metallofullerenes. The metal in TM@$C_{28}$ (TM = Sc$^-$, Ti, V$^+$, Ta$^+$) is positioned off-center and close to the carbon atom along the $C_3$ axis, giving rise to the $C_{3v}$ symmetry of this EMF. In contrast, [Zr@$C_{28}$] and [Nb@$C_{28}$]$^+$ exhibit the lowest symmetry ($C_{2v}$) due to the off-center positioning of the metal. Despite the Zr and Ti metals having the same d-electron configuration, Zr@$C_{28}$ ($C_{2v}$) and Ti@$C_{28}$ ($C_{3v}$) exhibit different symmetries and structures.

The Root Mean Square Deviation (RMSD) serves as an essential quantitative index for analyzing alterations in the symmetry and structural integrity of molecular frameworks. In Figure 2, visualized with the aid of VMD software [53], a structure from each symmetry class is presented, with concomitant structures co-aligned through strategic translation and rotation. Subsequent calculations of RMSD values between disparate structures (expressed in angstrom units) were conducted. RMSD values in Figure 2a,b primarily stem from the locomotion of the encapsulated transition metal atoms, while in Figure 2c, the RMSD is attributable to the dimensional augmentation of the carbon cage. Consequently, in the lower portion of Figure 2c, a line representation is adopted, where the central atoms of varying systems are overlaid in a spatial context, with lines in diverse hues representing the different structural configurations. This representational approach succinctly delineates the subtle disparities in atomic positioning and cage structural modifications resulting from the incorporation of distinct transition metals.

The RMSD values for the $C_{2v}$ symmetry systems with Zr and Nb$^+$ encapsulation are notably low (0.000 and 0.076, respectively), suggesting minor structural deviations due to the movement of the transition metals along the $C_2$ axis. For the $C_{3v}$ symmetry systems encompassing V$^+$, Sc$^-$, Ta$^+$, and Ti, there is an observed increment in RMSD values (0.000, 0.144, 0.164, and 0.181, respectively), indicating structural adaptations associated with the distribution of transition metals along different $C_3$ axes. In the $T_d$ symmetry systems containing Y$^+$, Hf, and La$^-$, the RMSD changes (0.000, 3.950, and 3.981, respectively) denote significant structural changes, principally due to the global expansion or contraction of the carbon cage. This behavior could be a direct consequence of the larger atomic radii of these encapsulated metals, which may enforce a more dramatic spatial rearrangement of

the carbon atoms, altering the fullerene cage dimensions. These observed RMSD variations between different metals underscore the influence of atomic size, electronic configuration, and oxidation state on the resulting symmetry and structural modifications within the fullerene cage. The complexity of these interactions suggests a nuanced interplay between metal properties and fullerene adaptability.

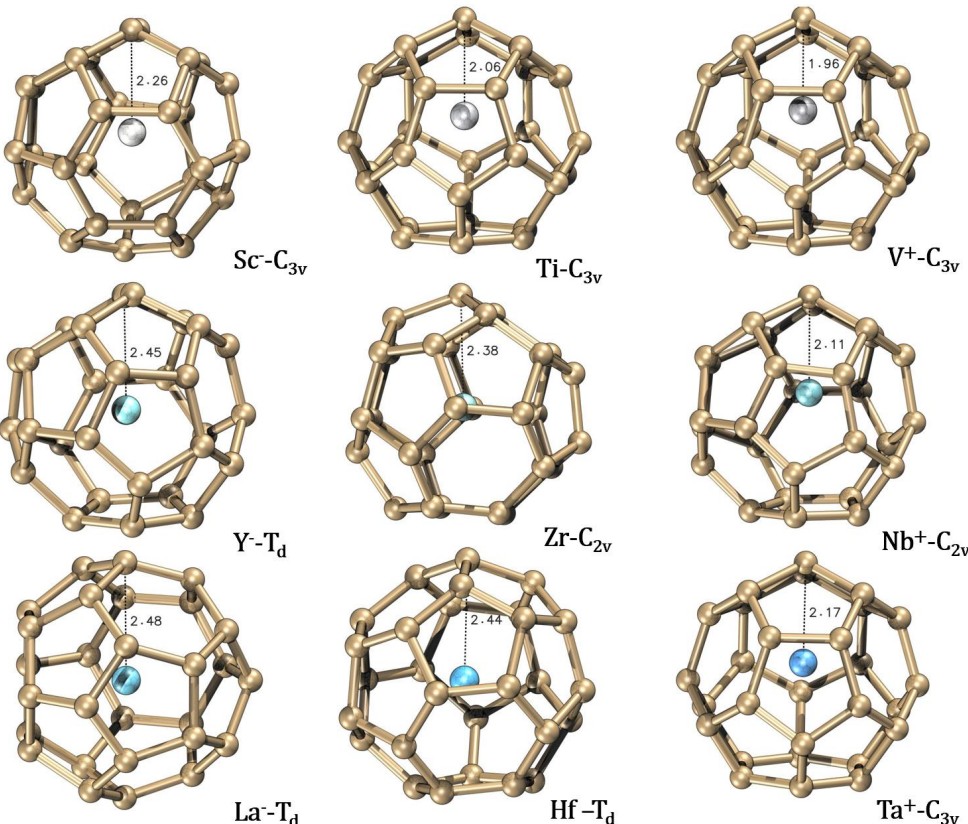

**Figure 1.** The optimized structures of TM@$C_{28}$ (TM= $Sc^-$, $Y^-$, $La^-$, Ti, Zr, Hf, $V^+$, $Nb^+$, $Ta^+$) with the shortest TM-C bond length.

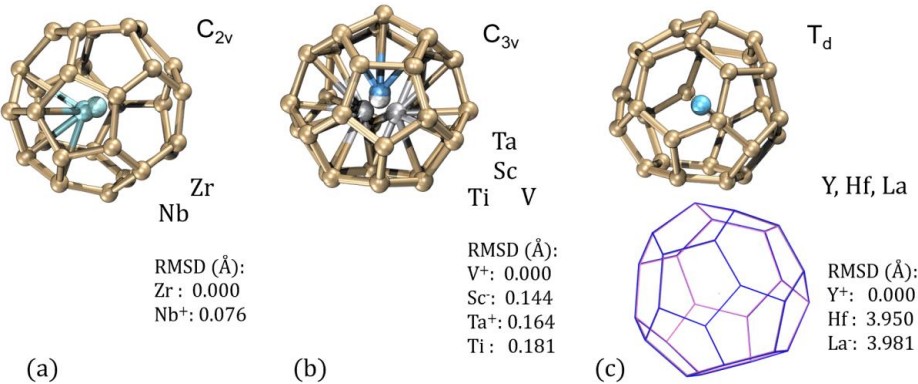

**Figure 2.** Visualization of transition metal-doped $C_{28}$ fullerenes exhibiting varied symmetry classes, analyzed using VMD software [53]. Panel (**a**) illustrates the $C_{2v}$ symmetric structure with Zr and $Nb^+$ encapsulation, panel (**b**) displays the $C_{3v}$ symmetric structure with $V^+$, $Sc^-$, $Ta^+$, and Ti encapsulation, and panel (**c**) represents the $T_d$ symmetric structure with $Y^+$, Hf, and $La^-$ encapsulation. The calculated RMSD values (in angstroms) highlight the structural deviations due to the specific movements of the embedded atoms and the expansion or contraction of the carbon cage. Panel (**c**) also features a line representation for clarity in structural differentiation, with superimposed central atoms from different systems and color-coded lines to distinguish between the structures.

The RMSD values derived from structural analysis provide a quantitative reflection of the divergence in geometric structures among different endohedral metallofullerenes. This divergence can be partially rationalized by applying the Hard and Soft Acids and Bases (HSAB) principle [54], which suggests a preferential interaction between "hard" acids and "hard" bases, as well as "soft" acids with "soft" bases. The HSAB principle is predicated on the notion that the chemical behavior of atoms and molecules during bond formation is influenced by their electronic properties, specifically the polarizability and the relative size of their orbitals.

Within the context of the periodic table, these properties are not constant; they demonstrate notable variation. For instance, as transition metal changes from titanium (Ti) to hafnium (Hf), there is an observed increase in the size of the metal ions. According to HSAB theory, this increment in size correlates with an increase in softness, which implies a greater capacity for deformation or polarization upon interaction with a base, which is the fullerene cage in this case. The softness of the metal ion can enhance the degree of distortion or expansion of the cage structure, as softer ions can form more malleable and adaptable bonds, accommodating themselves within the confines of the fullerene lattice.

Furthermore, this interaction is not merely a function of size or electronic softness; other chemical and physical parameters also play important roles. Steric effects, for example, pertain to the spatial arrangement of atoms and can drastically affect how a metal ion is incorporated into the fullerene cage. The specific nature of the metal–fullerene interaction, dictated by the unique chemistry of each metal, further contributes to the resultant geometric configuration. Additionally, the overall stability of the resulting complex is an aggregate effect of these individual interactions and properties.

It is also worth noting that while elements such as Ti, Zr, and Hf may share similar d-electron configurations, which suggests comparable chemical reactivity, the subtle differences in their ionic radii, softness, and other chemical properties can lead to distinct geometric structures within the fullerene framework. The HSAB concept provides a framework to understand these variations. A holistic understanding of the geometric forms adopted by these complexes necessitates an integrated approach, considering not only HSAB theory but also delving into quantum mechanical explanations, electron density distribution, and the dynamics of bond formation and breaking. Only by encompassing these multifaceted chemical and physical factors can one begin to fully elucidate the reasons behind the observed structural diversity in these endohedral metallofullerenes.

Table 1 presents the optimized shortest TM−C bond lengths of all nine EMFs. The TM−C distances range from 1.96 Å to 2.48 Å. It is not surprising that the shortest TM−C distance for the same group metal increases monotonically. For instance, the shortest TM−C distance for TM@$T_d$(2)-$C_{28}$ (TM = $Sc^-$, $Y^-$, $La^-$) increases from 2.26 to 2.48 Å. On the other hand, the TM−C distance decreases for the same period element. Particularly, the shortest V−C distance is only 1.96 Å. The calculations on the singlet–triplet splitting energy ($\Delta E_{S\text{-}T}$) revealed that all those values are larger than 1.04 eV, suggesting that the ground state for all these EMFs is a singlet.

**Table 1.** Symmetry (Sym.), the singlet–triplet splitting energy ($\Delta E_{S\text{-}T}$), the shortest TM−C bond length ($d_{TM-C}$) of TM@$C_{28}$ (TM = $Sc^-$, $Y^-$, $La^-$, Ti, Zr, Hf, $V^+$, $Nb^+$, $Ta^+$).

| EMFs | Sym. | $\Delta E_{S\text{-}T}$ (eV) | $d_{TM-C}$ (Å) |
|---|---|---|---|
| [Sc@$C_{28}$]$^-$ | ca. $C_{3v}$ | 1.87 | 2.26 |
| [Y@$C_{28}$]$^-$ | ca. $T_d$ | 1.98 | 2.45 |
| [La@$C_{28}$]$^-$ | ca. $T_d$ | 1.93 | 2.48 |
| [Ti@$C_{28}$] | ca. $C_{3v}$ | 1.56 | 2.06 |
| [Zr@$C_{28}$] | ca. $C_{2v}$ | 2.01 | 2.38 |
| [Hf@$C_{28}$] | ca. $T_d$ | 2.03 | 2.44 |
| [V@$C_{28}$]$^+$ | ca. $C_{3v}$ | 1.04 | 1.96 |
| [Nb@$C_{28}$]$^+$ | ca. $C_{2v}$ | 1.86 | 2.11 |
| [Ta@$C_{28}$]$^+$ | ca. $C_{3v}$ | 1.64 | 2.17 |

To assess the kinetic stabilities of the EMFs, Figure 3 presents the gap between the highest occupied molecular orbitals (HOMO) and the lowest unoccupied molecular orbitals (LUMO). Except Ti@$C_{28}$, [V@$C_{28}$]$^+$, and [Ta@$C_{28}$]$^+$, all the other EMFs hold a large HOMO–LUMO gap above 2.0 eV, indicating that these EMFs could be isolated experimentally. Particularly, both Zr@$C_{28}$ and Hf@$C_{28}$ have the highest HOMO–LUMO gap of 2.31 eV. On the other hand, the HOMO–LUMO gap of [V@$C_{28}$]$^+$ is as small as 1.35 eV, suggesting that this EMF is reactive and might not be isolated easily. Particularly, the HOMO–LUMO gap results are in agreement with the above the singlet–triplet splitting energy $\Delta E_{S\text{-}T}$ results; the larger the HOMO–LUMO gap, the larger $\Delta E_{S\text{-}T}$. Figure 4 presents the frontier molecular orbitals (FMOs) of three EMFs with different symmetries. Cage orbitals which are from the 2p atomic orbitals obviously contribute to all these FMOs. On the other hand, the metal orbitals show slight contributions. Particularly, the metal *d* orbitals show some contributions to the HOMO-1 and HOMO-2 of [Ta@$C_{28}$]$^+$ and HOMO of [La@$C_{28}$]$^-$.

To further analyze the electronic structures of TM@$C_{28}$, an NBO analysis was performed, and the results of the metals are shown in Table 2. The charges of metals range from $-0.18$ to $+1.81$. Although TM@$C_{28}$ (TM = Sc$^-$, Y$^-$, La$^-$) is negatively charged, the charge is mainly localized on the cage rather than on the metal, which is in line with the above molecular orbitals results. The highly positive charge ($+1.34$) of Y metal in [Y@$C_{28}$]$^-$ may come from the fact that the NBO analysis does not treat (n)p functions as valence orbitals of transition metals [55]. For the other six EMFs, the metal is positively charged, with the positive charge ranging from 0.53 to 1.81.

The formation energy of EMFs, which is defined by the energy difference from total energy minus the sum of neutral fullerene and neutral/charged transition metals, is important to assess the stability of EMFs. The formation energy of TM@$C_{28}$ was calculated and is shown in Table 2. All the formation energy falls in the range of $-5.91$ to $-10.63$ eV, indicating that these EMFs could form in the experiments. Particularly, Ti@$C_{28}$, which was observed experimentally, has a formation energy of $-8.18$ eV.

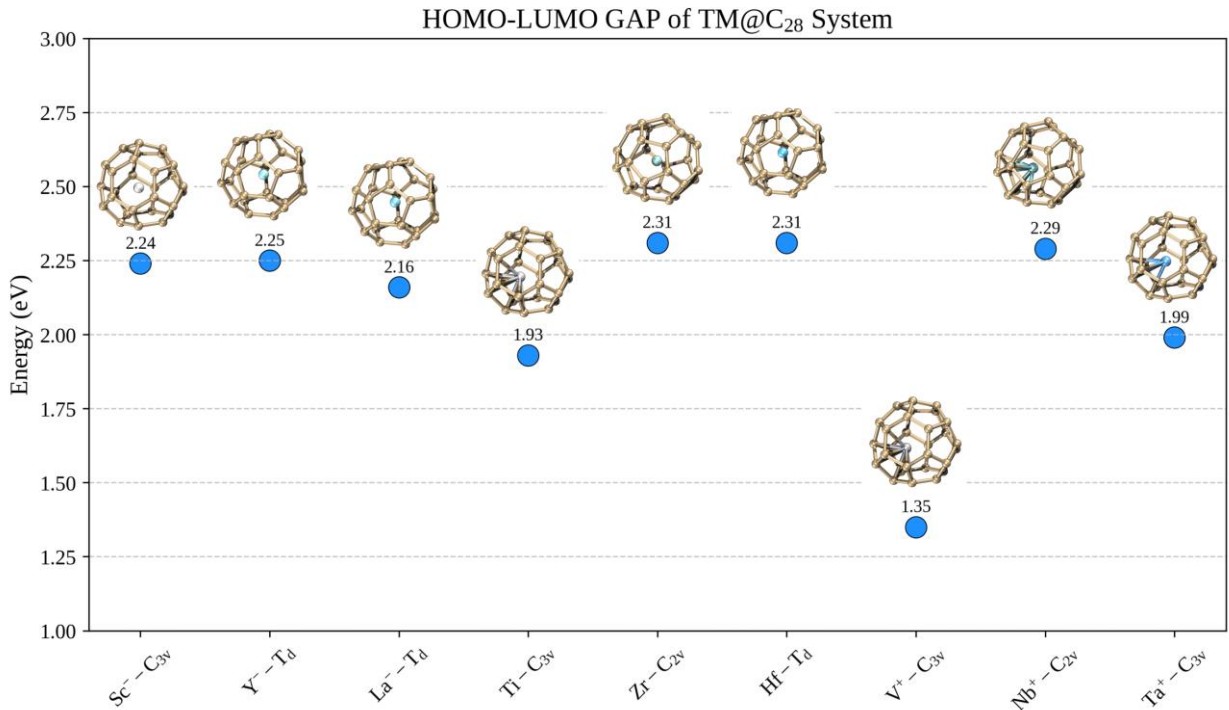

**Figure 3.** The HOMO–LUMO gap of TM@$C_{28}$ (TM = Sc$^-$, Y$^-$, La$^-$, Ti, Zr, Hf, V$^+$, Nb$^+$, Ta$^+$).

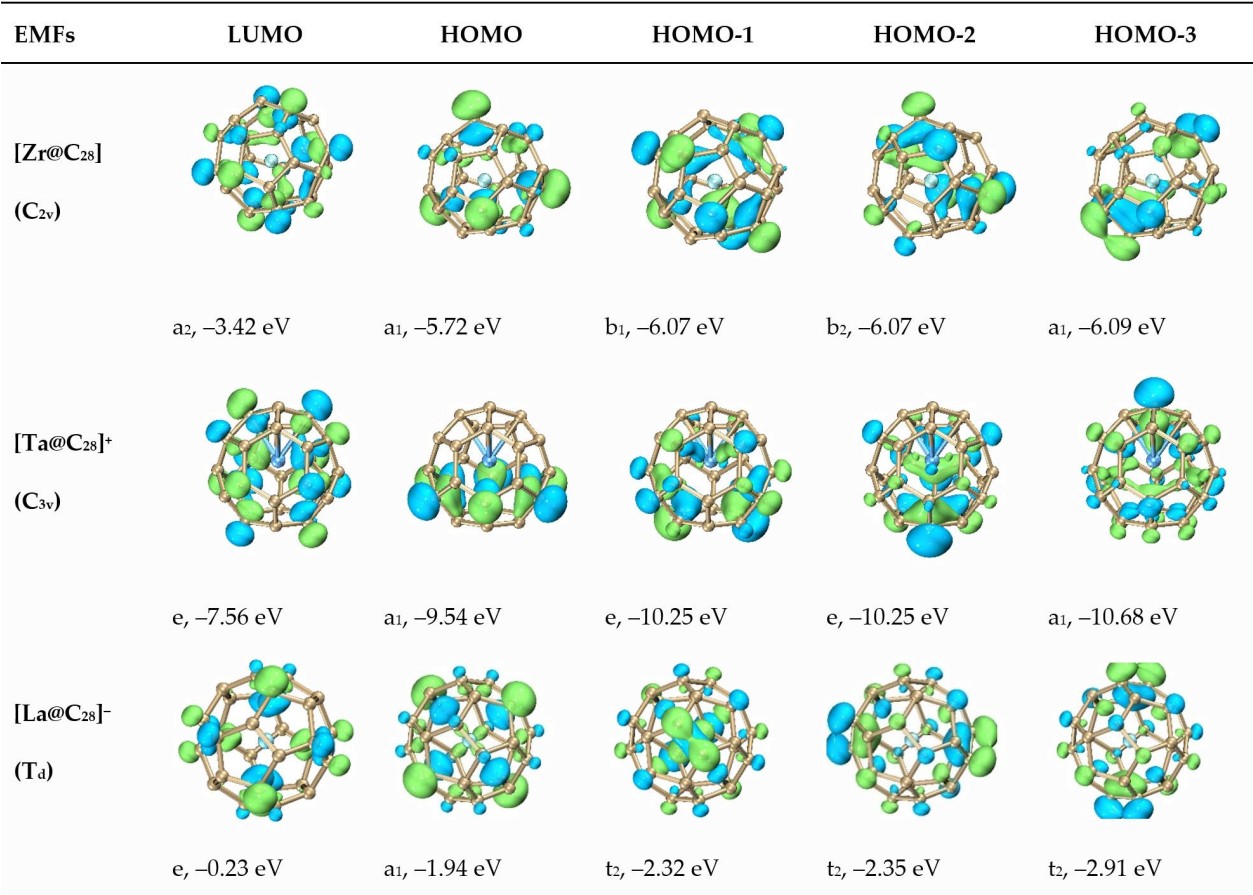

| EMFs | LUMO | HOMO | HOMO-1 | HOMO-2 | HOMO-3 |
|---|---|---|---|---|---|
| [Zr@C$_{28}$]<br>(C$_{2v}$) | a$_2$, −3.42 eV | a$_1$, −5.72 eV | b$_1$, −6.07 eV | b$_2$, −6.07 eV | a$_1$, −6.09 eV |
| [Ta@C$_{28}$]$^+$<br>(C$_{3v}$) | e, −7.56 eV | a$_1$, −9.54 eV | e, −10.25 eV | e, −10.25 eV | a$_1$, −10.68 eV |
| [La@C$_{28}$]$^-$<br>(T$_d$) | e, −0.23 eV | a$_1$, −1.94 eV | t$_2$, −2.32 eV | t$_2$, −2.35 eV | t$_2$, −2.91 eV |

**Figure 4.** Selected molecular orbitals (isovalue = 0.05) of [TM@C$_{28}$] at the PBE-D3(BJ)/def2-TZVPPD level.

**Table 2.** NBO partial charges, the formation energy from the carbon cage C$_{28}$ and transition metal atoms, and the first excitation energies and corresponding oscillator strengths of TM@C$_{28}$ (TM = Sc$^-$, Y$^-$, La$^-$, Ti, Zr, Hf, V$^+$, Nb$^+$, Ta$^+$). The T and Q represent the triplet and quintet states, respectively.

| EMFs (Singlet) | Fragment | Natural Charge (TM) | Formation Energy (eV) | Excitation Energies (eV) | Oscillator Strengths (nm) |
|---|---|---|---|---|---|
| [Sc@C$_{28}$]$^-$ | Sc$^-$(T) + C$_{28}$(Q) | −0.18 | −10.63 | 2.31 | 537.7 |
| [Y@C$_{28}$]$^-$ | Y$^-$(T) + C$_{28}$(Q) | 1.34 | −10.13 | 2.30 | 539.5 |
| [La@C$_{28}$]$^-$ | La$^-$(T) + C$_{28}$(Q) | 0.19 | −5.91 | 2.22 | 559.0 |
| [Ti@C$_{28}$] | Ti(Q) + C$_{28}$(Q) | 0.53 | −8.18 | 2.05 | 605.8 |
| [Zr@C$_{28}$] | Zr(Q) + C$_{28}$(Q) | 1.68 | −9.02 | 2.37 | 523.7 |
| [Hf@C$_{28}$] | Hf(T) + C$_{28}$(Q) | 1.64 | −8.66 | 2.37 | 523.9 |
| [V@C$_{28}$]$^+$ | V$^+$(Q) + C$_{28}$(Q) | 1.38 | −6.66 | 1.48 | 837.6 |
| [Nb@C$_{28}$]$^+$ | Nb$^+$(Q) + C$_{28}$(Q) | 1.33 | −8.00 | 2.38 | 521.5 |
| [Ta@C$_{28}$]$^+$ | Ta$^+$(Q) + C$_{28}$ (Q) | 1.81 | −8.47 | 2.13 | 582.9 |

The excitation energies and corresponding oscillator strengths, which are related to the electron excitation from the HOMO to LUMO, were calculated using time-dependent density functional theory (TDDFT), as shown in Table 2. The excitation energies range from 1.48 to 2.38 eV, with oscillator strengths ranging from 837.56 to 521.49 nm. The smaller the excitation energy, the larger the oscillator strength. The excitation energy also agrees well with the above HOMO–LUMO gap. The [V@C$_{28}$]$^+$ needs the smallest excitation energy.

Although the electronic structures of TM@C$_{28}$ like Ti@C$_{28}$ have been discussed in previous papers [13–16], the bonding nature between the metals and C$_{28}$ has not been analyzed.

Recently, we found that the EDA analysis could describe the bonding nature of EMFs [56] and Zintl clusters [57,58] well. Thus, to study the bonding nature between the encapsulated metal and fullerenes $C_{28}$, we performed the EDA analysis on TM@$C_{28}$. Because the EDA results strongly depend on how to divide a whole EMF into two interacting fragments, we first carried out the EDA analysis on the TM@$C_{28}$ by using different models. According to the criterion that the bonding model that has the smallest absolute orbital interactions $|\Delta E_{orb}|$ is usually determined as the best bonding model, the EDA results with the best bonding models are shown in Table 3, with the remaining models in the Supporting Information. As shown in Table 3, the bonding energy ranges from $-331.9$ kcal/mol to $-851.5$ kcal/mol, suggesting that all these EMFs are stable. The attractive interaction terms, including the Coulomb term $\Delta E_{elstat}$ and the orbital interaction term $\Delta E_{orb}$, compensate for the repulsive terms $\Delta E_{Pauli}$. In general, the Coulomb term $\Delta E_{elstat}$ contributes less than the orbital interaction term $\Delta E_{orb}$. Particularly, for [Nb@$C_{28}]^+$, the contribution from the orbital interaction term $\Delta E_{orb}$ to the total attractive energy is as high as 76.9%. The dispersion interactions contribute slightly to the total bonding strength with a range of $-2.8$ to $-6.9$ kcal/mol.

**Table 3.** EDA results (in kcal/mol) of the [TM@$C_{28}$] with $TM^{m+}$ and $C_{28}^{n-}$ as interacting fragments at the BP86-D3(BJ)/TZ2P+ level. The interaction energies ($\Delta E_{int}$), Pauli repulsion ($\Delta E_{Pauli}$), electrostatic interactions ($\Delta E_{elstat}$), orbital interactions ($\Delta E_{orb}$), and dispersion ($\Delta E_{disp}$) are presented with the percentages showing the contribution of orbital interactions or electrostatic attractions to the total attractive interactions.

| EMFs | [Sc@$C_{28}]^-$ | [Y@$C_{28}]^-$ | [La@$C_{28}]^-$ |
|---|---|---|---|
| Fragments | $Sc^+$ $(4s^03d^2)$ + $[C_{28}]^{2-}$ | $Y^+$ $(5s^04d^2)$ + $[C_{28}]^{2-}$ | $La^+$ $(6s^05d^2)$ + $[C_{28}]^{2-}$ |
| $\Delta E_{int}$ | $-552.3$ | $-361.4$ | $-308.7$ |
| $\Delta E_{Pauli}$ | $+472.3$ | $+752.4$ | $+1160.2$ |
| $\Delta E_{elstat}$ [a] | $-363.7$ (35.6%) | $-522.8$ (47.1%) | $-703.7$ (48.1%) |
| $\Delta E_{orb}$ [a] | $-658.1$ (64.4%) | $-588.3$ (52.9%) | $-760.8$ (51.9%) |
| $\Delta E_{disp}$ | $-2.8$ | $-2.8$ | $-4.3$ |
| **EMFs** | **[Ti@$C_{28}$]** | **[Zr@$C_{28}$]** | **[Hf@$C_{28}$]** |
| Fragments | $Ti^+$ $(4s^13d^2)$ + $[C_{28}]^-$ | $Zr^+$ $(5s^14d^2)$ + $[C_{28}]^-$ | $Hf^+$ $(6s^15d^2)$ + $[C_{28}]^-$ |
| $\Delta E_{int}$ | $-423.0$ | $-331.9$ | $-559.6$ |
| $\Delta E_{Pauli}$ | $+908.5$ | $+848.4$ | $+1053.6$ |
| $\Delta E_{elstat}$ [a] | $-464.5$ (35.0%) | $-485.7$ (41.3%) | $-585.9$ (36.4%) |
| $\Delta E_{orb}$ [a] | $-861.3$ (65.0%) | $-689.4$ (58.7%) | $-1024.4$ (63.6%) |
| $\Delta E_{disp}$ | $-5.7$ | $-5.2$ | $-2.8$ |
| **EMFs** | **[V@$C_{28}]^+$** | **[Nb@$C_{28}]^+$** | **[Ta@$C_{28}]^+$** |
| Fragments | $V^{2+}$ $(4s^03d^3)$ + $[C_{28}]^-$ | $Nb^{2+}$ $(5s^04d^3)$ + $[C_{28}]^-$ | $Ta^{2+}$ $(6s^05d^3)$ + $[C_{28}]^-$ |
| $\Delta E_{int}$ | $-851.5$ | $-508.0$ | $-622.1$ |
| $\Delta E_{Pauli}$ | $+831.1$ | $+1003.9$ | $+690.4$ |
| $\Delta E_{elstat}$ [a] | $-463.1$ (27.6%) | $-348.2$ (23.1%) | $-388.8$ (29.7%) |
| $\Delta E_{orb}$ [a] | $-1214.1$ (72.4%) | $-1156.9$ (76.9%) | $-918.8$ (70.3%) |
| $\Delta E_{disp}$ | $-5.4$ | $-6.9$ | $-4.9$ |

[a] The percentages corresponding to the total attractive interactions ($\Delta E_{elstat} + \Delta E_{orb}$) are in parentheses.

The HOMO–LUMO gap of [V@$C_{28}]^+$ is as small as 1.35 eV, which is less than the average HOMO–LUMO gap (2.09 eV) in the nine EMFs, suggesting the relatively lower kinetic stability of [V@$C_{28}]^+$. Previous studies have shown that electronegative organic groups like trifluoromethyl can enhance the stability of EMFs by increasing the HOMO–LUMO gap [59,60]. In these previous studies, La@$C_{60}$, which has an open-shell structure and is difficult to isolate experimentally, is theoretically predicted to be able to be stabilized by introducing $CF_3$ groups [56]. Then, this theoretical proposal was verified by the experimental

single-crystal structure of the CF$_3$-adduct of La@C$_{60}$ [57]. In the present study, we studied the addition of the trifluoromethyl group to [V@C$_{28}$]$^+$ to enhance its stability and facilitate its future experimental isolation.

We focused on the single addition and double addition of the trifluoromethyl group to [V@C$_{28}$]$^+$. For the lowest-energy [V@C$_{28}$]$^+$ (C$_{3v}$), we tried all 28 configurations (C$_{28}^1$) of single CF$_3$ addition and 378 configurations (C$_{28}^2$) of two CF$_3$ additions at the PBE-D3(BJ)/def2-SVP level. After optimizations, only 7 structures and 51 structures exist for [V@C$_{28}$]$^+$-CF$_3$ and [V@C$_{28}$]$^+$-(CF$_3$)$_2$, respectively. Figure 5 shows the seven possible adducts of [V@C$_{28}$]$^+$-CF$_3$ with the relative reaction energies and HOMO–LUMO gaps, and the remaining results are shown in Supporting Information. The two lowest-energy structures of [V@C$_{28}$]$^+$-CF$_3$ have almost the same reaction energy of 68.3 kcal/mol and 69.2 kcal/mol, respectively. The HOMO–LUMO gaps of these two structures are 1.72 and 1.84 eV, obviously higher than that of [V@C$_{28}$]$^+$.

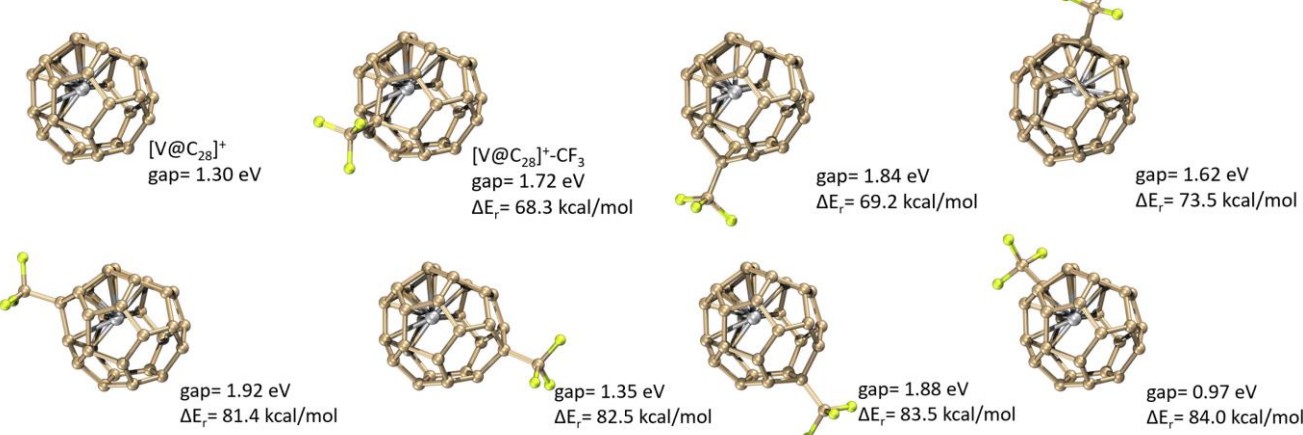

**Figure 5.** The HOMO–LUMO gaps and reaction energies of [V@C$_{28}$]$^+$ and [V@C$_{28}$]$^+$-CF$_3$ isomers at the PBE-D3(BJ)/def2-SVP level.

Further addition of one more CF$_3$ group to [V@C$_{28}$]$^+$-CF$_3$ gives rise to lots of possible [V@C$_{28}$]$^+$-(CF$_3$)$_2$ isomers. Figure 6 only presents four isomers with relative energy between [V@C$_{28}$]$^+$ and two CF$_3$ groups. The lowest-energy isomer has the two CF$_3$ group additions on the two carbon atoms connected by one C−C bond with a reaction energy of 152.5 kcal/mol. The HOMO–LUMO gap then increases to 2.18 eV. The reaction energy of the second isomer is only 1.20 kcal/mol higher than that of the first one, but its HOMO–LUMO gap is only 1.78 eV, indicating its lower kinetic stability. Therefore, it is clear from the above discussion that the addition of CF$_3$ groups to [V@C$_{28}$]$^+$ will increase its stability, which will assist future experimental isolation of [V@C$_{28}$]$^+$-(CF$_3$)$_2$.

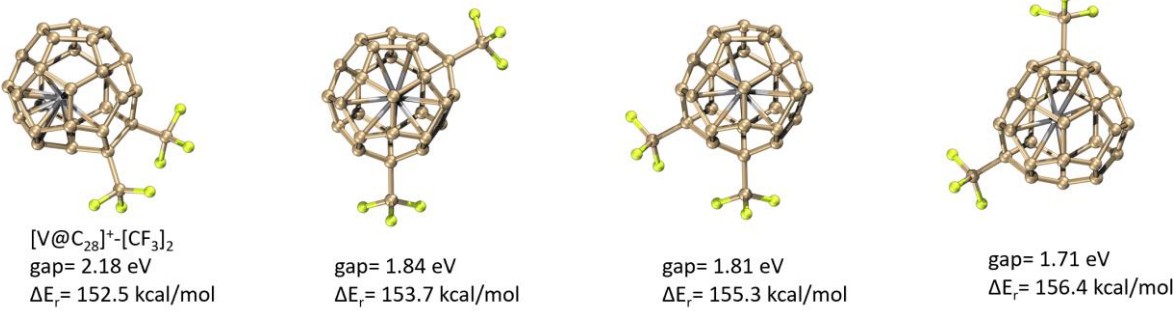

**Figure 6.** The HOMO–LUMO gaps and reaction energies of low-energy [V@C$_{28}$]$^+$-(CF$_3$)$_2$ isomers at the PBE-D3(BJ)/def2-SVP level.

## 4. Conclusions

By using quantum chemical calculations, we studied the geometries, electronic structures, bonding properties, and stability strategy of endohedral metallofullerenes $TM@C_{28}$ ($TM = Sc^-$, $Y^-$, $La^-$, $Ti$, $Zr$, $Hf$, $V^+$, $Nb^+$, $Ta^+$). $TM@C_{28}$ has three different structures with $C_{2v}$, $C_{3v}$, and $T_d$ symmetries. The HOMO–LUMO gap ranges from 1.35 eV to 2.31 eV, with $[V@C_{28}]^+$ having the lowest HOMO–LUMO gap of 1.35 eV. The trend of singlet–triplet splitting energy and first excitation energy is in line with the HOMO–LUMO gap. The frontier molecular orbitals analysis revealed that the molecular orbitals are mainly from fullerene cage orbitals, with a slight contribution from the encapsulated metals. The bonding interactions between encapsulated metal and fullerene cages are dominated by the Coulomb term $\Delta E_{elstat}$ and the orbital interaction term $\Delta E_{orb}$, in which the orbital interaction term $\Delta E_{orb}$ contributes more than the Coulomb term $\Delta E_{elstat}$. The addition of one or two $CF_3$ groups to $[V@C_{28}]^+$ could increase the HOMO–LUMO gap, giving rise to the possible isolation of $[V@C_{28}]^+$.

**Supplementary Materials:** The following supporting information can be downloaded at https://www.mdpi.com/article/10.3390/inorganics12020040/s1, Figure S1: The HOMO-LUMO GAP after attaching a CF3 group to [V@C28]+ and the energy differences between different isomers, at the PBE-D3(BJ)/def2-SVP level.; Table S1: The atomic (ionic) energies (in eV) calculated using PBE-D3(BJ)/def2-TZVPPD.; Figure S2: After the addition of two CF3 groups, the HOMO-LUMO GAP and the energy relationships among different isomers, at the PBE-D3(BJ)/def2-SVP level; Table S2: The EDA results (in kcal/mol) of the $[TM@C_{28}]$ with $TM^{m+}$ and $C_{28}^{n-}$ as interacting fragments at the BP86-D3(BJ)/TZ2P+ level.

**Author Contributions:** For the author contributions in our manuscript: D.L. was responsible for the majority of the computations, analysis, and original draft preparation. Y.S. contributed to parts of the data discussion and analysis. T.Y., as the corresponding author, provided project management, funding acquisition, supervision, and was involved in the review and editing of the manuscript. All authors have read and agreed to the published version of the manuscript.

**Funding:** This research was funded by the National Natural Science Foundation of China, grant number 12274337.

**Data Availability Statement:** We emphasize open data sharing and hereby provide the following data availability statement: All research data have been meticulously recorded and are stored in a trusted data repository. Some essential data are attached in the Supplementary Materials of this paper for reference. We encourage interested researchers to contact us for further details on data availability.

**Acknowledgments:** The authors gratefully acknowledge the financial support from the National Natural Science Foundation of China (Grant No. 12274337).

**Conflicts of Interest:** The authors declare no conflict of interest.

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
