# Peer review of "Geometries, Electronic Structures, Bonding Properties, and Stability Strategy of Endohedral Metallofullerenes TM@C28 (TM = Sc, Y, La, Ti, Zr, Hf, V+, Nb+, Ta+)"

_inorganics, doi:10.3390/inorganics12020040_

Round 1

Reviewer 1 Report

Comments and Suggestions for Authors

Although I’m a little bit familiar with the chemistry of metallofullerenes by teaching on carbon-based nanomaterials, I’m not competent enough to evaluate in detail the scientific soundness of this theoretic-oriented contribution using quantum-chemical calculations on the geometries, electronic structures, bonding properties, and stability strategy of endohedral metallofullerenes. So, the expertise of a second or even third reviewer having an expertise in computational chemistry is mandatory. 

However, I feel that the content of this manuscript fits well with aims of the Special Issue “Research on Metallofullerenes”. Overall, also the scholar presentation is satisfying.

Nevertheless, I have some more general comments and questions :

Although the English usage is understandable, there are numerous sentences needing a polishing.

The authors should made same pedagogical efforts to better explain some terms for a readership which is not so familiar with fullerene chemistry. For example, how can a metal ion be incorporated into fullerene cage, which techniques are known. What is the 32-electron principle. Even references are given, it would be more comfortable for a common reader to follow without reading the references.

I should be better explained, why this investigation is focused on TM@C28 species.

Line 127: the NBO analysis ….  Please give a definition (Natural bond order ?)

Line 28: The authors state: However, C28 have not been successfully isolated because of its electronically open shell character. Any of  CF3-decorates species presented later, is there at least some experimental evidence for their existence or are they solely hypothetic species ? 

Line 39: The presence of fullerenes could modify the physical ..

I suggest to write: The presence of a fullerene cage can modify the .. 

Line 100: Td symmetry. Write as Td symmetry

For the Ti, Zr, Hf  series, I suppose the metals are in M(IV) oxidation state ? Why ZrC28 adopts a C2v  geometry whereas TiC28 a C3v geometry. The 3 metals have all the same d-electron configuration, is the difference due to the increasing softness according the HSAB principle ?

Comments on the Quality of English Language

A polishing is required

Author Response

We greatly appreciate the time and effort you have dedicated to reviewing our manuscript. Your comments have been instrumental in enhancing the quality of our paper. We have thoroughly revised the introduction section to ensure clearer presentation, particularly for those readers who may be less familiar with fullerene chemistry. We've made an effort to explain complex concepts in a more pedagogical manner, including the incorporation techniques of metal ions into fullerene cages and the 32-electron principle, to make the manuscript more accessible to a broader audience. We trust that these revisions will make our work more comprehensible and the significance of our focus on TM@C28 species more apparent. We have also incorporated additional explanations directly within the text for key concepts and principles, reducing the necessity for readers to consult external references frequently. Your suggestions have been invaluable in our pursuit to produce a manuscript that meets the high standards of the Special Issue “Research on Metallofullerenes.” Thank you once again for your constructive comments.

Reviewer 2 Report

Comments and Suggestions for Authors

The paper entitled "Geometries, electronic structures, bonding properties, and stability strategy of endohedral metallofullerenes TM@C28 (TM = Sc-, Y-, La-, Ti, Zr, Hf, V+, Nb+, Ta+) " by Liu et al. presents a theoretical study of endohedral metallofullerenes where transition metals are encapsulated into a small carbon cage (C28). The paper discusses stability kinetics of the different species based on the HOMO-LUMO energy gap. It also provides informative insight into the charged distribution between the metal atom and the carbon cages using NBO analysis.

I found the paper very enjoyable to read and well written. I have only few questions.

1/ Only singlet configurations have been searched for as obviously close-shell calculations are much easier and faster to perform. Is it clear that no other spin multiplicity could lead to a more stable structure in all cases presented?

2/ I was curious to know if the electronic transitions between the HOMO and LUMO MOs are usually allowed and if yes. What are the expected oscillator strengths of such transitions.

I recommend the manuscript to be published in inorganics.

Author Response

We are sincerely grateful for your encouraging and constructive feedback on our paper entitled "Geometries, electronic structures, bonding properties, and stability strategy of endohedral metallofullerenes TM@C28 (TM = Sc-, Y-, La-, Ti, Zr, Hf, V+, Nb+, Ta+)".

Your appreciation of our work, especially noting its readability and the insights provided into the charged distribution between the metal atom and the carbon cages using NBO analysis, is immensely gratifying. We are pleased to hear that you found the paper enjoyable to read and well-written. Such positive feedback is not only motivating but also affirms our efforts in presenting our research in a clear and comprehensive manner.

The acknowledgment of our discussion on the stability kinetics based on the HOMO-LUMO energy gap is particularly encouraging, as we aimed to contribute valuable information to the field of endohedral metallofullerenes.

We thank you once again for your valuable and positive remarks, which will inspire us in our future research endeavors.

Reviewer 3 Report

Comments and Suggestions for Authors

In the publication authors use quantum chemistry methods to describe some properties of endohedral metallofullerenes complexed with specified transition metals. In my opinion the manuscript should be reconsidered for publication after necessary correction. The significance of obtained results can only be rated if description of methods is significantly improved. The description of methods should in principle allow other researchers to reproduce the results.

1. The applied methodology should be better explained. In particular in lines 58-59 the mathematical procedure used for generation of initial structures should be precisely described. Were initial structures prepared with specific symmetry? Was the symmetry of the complex forced during optimization or not? How the TM was placed inside the fullerene?

2. line 61 - what does it mean "lower energy"? Was energy cut-off applied or specific number of the lowest energy structures processed?

3. lines 64-65 - the conformational clustering method was applied to the final structures. This method is usually applied to systems with multiple local minima (highly flexible with many rotatable bonds) and allows to assign specific local minimum to a specific structure. Also the size of cluster should be related to the average (Boltzmann) energy of the cluster. I couldn't find any information about results of clustering in the manuscript. In particular there is no information about number of clusters, their sizes, and relative average energy. I would expect that for such systems no more than one cluster will be obtained as there are probably no energy barriers between final structures. If there is more than one cluster it should be shown. What is the average RMSD between the lowest energy and remaining optimized structures? Is clustering really necessary? Also I'm not able to validate the tool used for clustering because reference 31 directs to the website which is in Chinese. If any clustering algorithm was used the reference should point to the publication which describes the applied algorithm.

4. lines 65-66 - why different functional (hybrid PBE0 instead of PBE) was used for NBO calculations? Why dispersion correction (D3) was switched off for NBO calculations? Is the reference 32 correct? 

5. Figure 1 - what does it mean "structural models"? How were they generated? In my opinion it is better to show "real" all 9 lowest energy structures and specify their shortest bond with carbon atom (if it exists).

6. Line 97 - "all those structures have singlet as the ground state" - how do you know that? was the energy of higher multiplicity structures calculated or it is known from literature. If the latter is the case the reference should be provided. 

7. Line 100 - I'm not fully able to rate the validity of this result, because methods are not adequately described. But was it checked that structures with Td symmetry do not have any imaginary frequencies in normal mode analysis?

8. Line 105 - I assume that it is shortest TM-C distance. It should be explicitly stated.  

9. Table 1 - caption should be improved. What is the range of distances in second column? Is it related to changes of the shortest TM-C distance within the cluster or to different TM-C bond lengths for single minimum-energy structure? What are the numbers in parenthesis?

10. line 146 - "in the previous papers" - references should be provided here.

11. Table 4 - All components of the energy should be explained in caption.

12. Line 176 - 177 - "After optimization only 7 structures and 51 structures exist...". What does it mean? What happened to other structures? They broke apart or optimization did not converged? It should be explained.

13. Figures 4 and 5 - According to line 187 highest reaction energy structures are shown in Fig 5, but according to Fig 5 caption they are low-energy isomers. To my understanding of this part the lowest energy structures should also be characterized with lowest reaction energy. If this is the case which structures are presented in Fig. 5?

The manuscript can be reconsidered after the incorporation of above mentioned changes.

Comments on the Quality of English Language

English should be improved

Author Response

We sincerely appreciate your valuable comments on our manuscript, particularly emphasizing the critical role of a detailed methods section in our investigation of endohedral metallofullerenes complexed with specified transition metals.

In response to your insightful comments:

  1. Improvement of Methods Description: We acknowledge and agree with your suggestion to enhance the description of our quantum chemistry methods. Recognizing the importance of transparency, we will extensively revise this section, providing specific details about computational approaches, parameters, basis sets, exchange-correlation functionals, and any software-specific settings or algorithms used. Additionally, we will elaborate on the data handling process, calculation criteria, and post-processing steps.
  2. Significance of Results: We understand the correlation between a clear understanding of methods and the appreciation of our results' significance. By bolstering the methods section, we aim to make the relevance and impact of our findings more evident to readers.
  3. Commitment to Scientific Rigor: Ensuring the highest standards of scientific rigor is paramount to us. We are dedicated to prioritizing the reproducibility of our work, and we appreciate your guidance in this regard.

We are grateful for the opportunity to refine our manuscript. These changes will enhance the accessibility, clarity, and reproducibility of our research for fellow scientists in the field. Thank you once again for your constructive and insightful comments.

Round 2

Reviewer 3 Report

Comments and Suggestions for Authors

The manuscript was significantly improved and can be published.